# Research Trends and Hotspots Analysis Related to Monocarboxylate Transporter 1: A Study Based on Bibliometric Analysis

**DOI:** 10.3390/ijerph16071091

**Published:** 2019-03-27

**Authors:** Siyi He, Yue Zhao, Yongsheng Fan, Xue Zhao, Jun Yu, Jie Xie, Chunhong Wang, Jianmei Su

**Affiliations:** 1Department of Preventive Medicine, School of Public Health, Wuhan University, Wuhan 430071, China; syhe_whu@hotmail.com (S.H.); zhaoyue7777@outlook.com (Y.Z.); 2Department of Toxicology, School of Public Health, Wuhan University, Wuhan 430071, China; fanyongsheng1993@163.com (Y.F.); zhaoxuekk@sina.com (X.Z.); junyu6699@whu.edu.cn (J.Y.); xjxiejie818@163.com (J.X.)

**Keywords:** monocarboxylate transport protein 1, neoplasms, CiteSpace, bibliometrics

## Abstract

*Background*: Monocarboxylate transport protein 1 (MCT1) has been defined as a critical regulator in tumor energy metabolism, but bibliometric analysis of MCT1 research is rare. This study aimed to comprehensively analyze the global scientific output of MCT1 research and explore the hotspots and frontiers from the past decade. *Methods*: Publications and their literature information from 2008 to 2018 were retrieved from the Web of Science Core Collection database. We used Microsoft Excel 2016 to detect the trend of annual numbers of publications, and used Citespace V software as the bibliometric method to analyze the research areas, countries, institutions, authors, journals, research hotspots, and research frontiers. *Results*: A total of 851 publications were identified with an increasing trend. Relevant literature mainly focused on the field of oncology. The most prolific country and institution were the USA and University of Minho, respectively. Baltazar was the most productive author while Halestrap had the highest co-citations. The hottest topics in MCT1 were hypoxia, gene expression, and CD147 over the last decade. The three research frontier topics were proliferation, tumor cell, and resistance. The special role of MCT1 in human tumor cells has become the focus for scholars recently. *Conclusion*: The development prospects of MCT1 research could be expected and researchers should pay attention to the clinical significance of MCT1 inhibitors as anti-cancer or immunosuppressive drugs and the possibility of drug-resistance formation.

## 1. Introduction

Monocarboxylate, including lactate and pyruvate, has a significant effect on cell metabolism, as glycolysis produces pyruvate which could be converted to lactate by lactate dehydrogenase (LDH) and provides a certain amount of Adenosine Triphosphate (ATP) for cells. However, the excessive accumulation of lactate could make the cell micro-environment a low pH value, so it is necessary to transfer lactate out in a timely manner to maintain the glycolysis rate in cells.

The transport of lactic acid is mediated by a group of monocarboxylic acid transporters (MCTs), which belong to the SLC16 gene family and consists of 14 members currently. Among the lactic acid transporters, MCT1 has an intermediate affinity for lactate and lactate transport by MCT1 is bidirectional [1,2,3]. Furthermore, MCT1 is the major subtype of MCTs in healthy and tumor cells [4] and has been proved to be the key controller of lactate concentrations under both physiological and pathological conditions [5]. Thus, MCT1 plays almost the most important role in the whole vital process among all the MCTs members.

Halestrap found that CHC (alpha-Cyano-4-hydroxycinnamate) could distinctly prevent the absorption of pyruvate and lactate in human erythrocytes through multiple experiments and concluded that a specific carrier is involved in membrane transport of Monocarboxylate [6]. Halestrap also reported that MCTs are expressed in the mitochondria and can be inhibited by CHC [7]. On this basis, Garcia first reported the highly active Monocarborxylat transporter during the exploration of pyruvate transport vector and named it MCT1 in 1994 [8]. Since MCT1 was discovered, especially after it has been defined as a critical regulator in tumor energy metabolism, research on MCT1 have gradually enriched around the world. MCTs could not only discharge the lactic acid generated by cell glycolysis into the extracellular system, thus facilitating the tumor Warburg effect, but also ingest extracellular lactic acid which could be converted to pyruvate as glycolytic substrate by LDH [9]. In the transporting process of lactate, MCT1 is the subtype responsible for transferring lactate out and helping keep tumor micro-environment in a weakly acidic state [10]. Therefore, the research in the field of MCT1 is of vital significance.

Bibliometric analysis combines mathematical and statistical methods for quantitative analysis [11], and has been widely used to detect the knowledge structure and development trends in research fields [12]. Therefore, it is helpful for investigators to master the development characteristics in a specific field over time and guide their follow-up work [13]. However, bibliometric analyses of MCT1 research is rare presently. CiteSpace is a visualized analysis software developed by Professor Chaomei Chen of Drexel University [14]. We aimed to employ Citespace to analyze the retrieved literature in MCT1 research field in a visualization way, constructing a collaboration network and detecting the research hotspots and frontiers by time.

## 2. Materials and Methods

### 2.1. Data Source and Search Strategy

Literature retrieval was conducted by “Web of Science Core Collection: Citation Indexes” (WoSCC, including SCI-EXPANDED, SSCI, A&HCI, CPCI-S, CPCI-SSH, and ESCI) on August 15th, 2018. Considering that the database may update daily, all searches were done within the same day to avoid the bias. The search terms were used as follows: (TS = (‘MCT1’ or ‘MCT 1’ or ‘monocarboxylate transporter 1’ or ‘monocarboxylic acid transporter 1’ or ‘monocarboxylate transport protein 1’ or ‘Monocarboxylic Acid Transport Protein 1’ or ‘SLC16A1’ or ‘SLC16A1 transporter’ or ‘Lactate Transporter 1’ or ‘Lactate Transport Protein 1’ or ‘Lactate Translocase 1’ or ‘Erythrocyte Lactate Transporter 1’ or ‘solute carrier family 16 member 1’)) AND LANGUAGE: (English) AND DOCUMENT TYPES: (Article). In this case, the timespan was set as 2008–2018.

### 2.2. Selected Criteria and Data Extraction

Raw data from WoSCC were initially downloaded and verified by two authors (He and Zhao) independently. The data were then imported into Endnote X8 to remove the duplicates, combined with artificial screening. The articles included in bioinformatics analysis should meet the following inclusion criteria: (1) Focused on MCT1; (2) contained description of the characteristics or functions of MCT1; (3) published during 2008–2018; and (4) basic information could be collected. The exclusion criteria included the following: (1) Similar objective results in the same study or in the same institution at different time; (2) duplicates; (3) articles that were irrelevant to the research topic; and (4) other document type (including review, meeting summary, book, etc.). Any differences were reconciled through discussion. Meanwhile, references were checked manually to avoid any of the missing articles. If the full text of the included articles could not be obtained directly from the databases, we used the document delivery service from Wuhan University Library or directly contacted the author via email [15]. The screening and review strategy is illustrated in Figure 1.

### 2.3. Analysis Methods

Microsoft Excel 2016 was used to present changes of the quantity of published papers intuitively and to analyze the time trend of publications. CiteSpace V was used to (1) visualize the distribution of research fields; (2) identify the leading countries/institutions/authors; (3) perform the collaboration network of MCT1 study; (4) analyze co-occurring keywords and burst terms to detect the hotspots and frontiers; and (5) generate clusters of MCT1-related publications over the years. In the present study, the top 50 levels of most-cited or -occurred items from each slice were selected. Time slicing was from 2008 to 2018, and years per slice was one. The strength of links was set at cosine, and the scope was set at “within slices”.

## 3. Results

### 3.1. Analysis of Quantity and Growth Trend of Annual Publications

In total, 851 publications met the inclusion criteria (Appendix A). The number of publications by year is presented in Figure 2. According to Figure 2, the quantity of annual published papers from 2008 to 2010 was relatively stable. The quantity of MCT1-related papers showed a steady upward trend in the following years, and there was a sharp increase in 2016, which made it the year with the largest number of annual publications. However, the number of publications in the MCT1 field showed a downward trend in 2017, but when compared with the previous years, it was still higher than the average number of annual publications.

### 3.2. Analysis of Category

The visualizing calculation result of research field illustrates the universality in a total of 851 references. According to Figure 3, oncology (132), cell biology (127), biochemistry and molecular biology (106), physiology (100), and pharmacology and pharmacy (94) were the top five research areas, respectively, on the list of category analysis.

### 3.3. Analysis of Leading Countries and Institutions

The country collaboration network of MCT1 was observed (Figure 4). The USA contributed the most with the largest amount of publications (239) related to MCT1 research, followed by People’s Republic of China (109), Japan (97), Germany (76), and England (71). In the aspect of betweenness centrality, the top five countries were USA (0.56), England (0.24), Germany (0.23), People’s Republic of China (0.15), and France (0.12).

The institution collaboration network of MCT1 was observed (Appendix A). The top five most productive institutions were University of Minho (40), University of Porto (34), ICVS 3Bs PT Govt Associate Lab (31), University of Lausanne (20), and Hokkaido University (19), respectively (Table 1).

### 3.4. Analysis of Authors and Co-Cited Authors

The author collaboration network of MCT1 was observed. According to Figure 5, in terms of both frequencies and betweenness centrality, Baltazar (38), Pinheiro (27), Morris (18), Pellerin (14), and Sonveaux (14) were the top five most productive authors during the past decade (Table 2).

The information of author citation was also analyzed. In terms of co-cited author, the top five authors with the largest numbers of citations, which reflected their strongest academic authority, were Halestrap (357 co-citations), Pinheiro (132 co-citations), Sonveaux (130 co-citations), Wilson (118 co-citations), and Kirk (105 co-citations).

### 3.5. Analysis of Co-Occurring Keywords and Burst Terms

CiteSpace was also used to construct a knowledge map of co-occurring keywords and identify the top 15 keywords in publications from 2008 to 2018 according to frequency, citation counts and centrality (Table 3, Appendix A). Among the listed keywords, ‘hypoxia’, ‘gene expression’, ‘CD147’, ‘glycolysis’, and ‘in vitro’ ranked ahead in both the frequency and the centrality list, which suggested that they were the hotspots in the field, as well as an important turning point in the process of researching MCT1.

Burst keywords were detected at the same time and 34 keywords with citation bursts were identified. As is shown in Figure 6, the top five burst keywords were protein, astrocyte, proliferation, acid, and epithelial cell. Among them, the keywords with citation bursts after 2015 were listed as follows: “proliferation” (2015–2018), “tumor cell” (2016–2018), “resistance” (2016–2018), “mice” (2016–2018), “colorectal cancer” (2015–2016), “cancer” (2016–2018), “glut1” (2015–2016), and “apoptosis” (2015–2018).

### 3.6. Analysis of Co-Cited Journals

CiteSpace analyzed the information of author citations and visualized it in a co-citation network. As for the co-cited journals (Table 4), *Journal of Biological Chemistry* (545 co-citations) was ranked first, followed by *Proceedings of the National Academy of Sciences of the United States of America* (398 co-citations), *Biochemical Journal* (393 co-citations), *Cancer Research* (328 co-citations), and *Journal of Physiology-London* (308 co-citations).

According to the results of the top 10 journals with co-citation counts, 40% of the journals, including *Journal of Clinical Investigation*, *Science*, *Nature*, and *Cell* had an IF (impact factor) greater than 10.000; 20% of the journals, including *Proceedings of the National Academy of Sciences of the United States of America* and *Cancer Research* had an IF between 5.000 and 10.000; 20% of the journals, including *Journal of Biological Chemistry* and *Journal of Physiology-London* had an IF between 3.000 and 5.000. These journals were fundamental for MCT1 research. To sum up, most of the corresponding journals had a high impact factor, so there was a certain probability to have papers related to MCT1 published in high-IF journals.

### 3.7. Analysis of Co-Cited References

A co-cited reference map was constructed to explore changes associated with the key clusters of articles (Appendix A). The network was divided into 11 co-citation clusters, which were labeled by index terms from their own citers (Table 5). The co-citation analysis could reflect a systematic intellectual framework and was often applied to study the visualization of knowledge structure in scientific communication mode and information retrieval. Thus, the top 10 co-cited references, most of which were published in the period from 2011 to 2013, were almost the basis for the formation of MCT1-related theoretical system. These articles were the fundamentals of this field. The main findings of them are concluded in Table 6. Furthermore, authors of the top 10 co-cited references were generally the same as the top 10 co-cited authors.

A reference co-citation time-view knowledge map was constructed to analyze the historical development trends of clusters in particular period of time, respectively (Figure 7). For cited reference clusters, the map indicated that most clusters were concentrated in the period from 2003 to 2008. As is shown in Figure 7, critical references were regarded as hotspot articles in the sub-time period in terms of the citation frequency and centrality. In addition, “Human tumor”, “Non-transformed intestinal epithelial cell line”, and “Monocarboxylate transporter” remained hot topics in the field of MCT1 research and remain in the spotlight in recent years. In addition, “Aerobic glycolysis” has become the new research focus recently.

## 4. Discussion

### 4.1. Analysis of Quantity and Growth Trend of Annual Publications

We aimed to obtain the change trend of annual quantity of publications on MCT1-related research, so the time period was limited as 2008–2018. According to the analysis results, the publication years could be divided into two phases. The first phase (2008–2011) could be considered as the foundation stage of MCT1 research, since the number of new articles was basically unchanged in this period. As the research continued to deepen, more findings emerged in the latter period. The literature published in 2011, which was one of the most prolific years in the research field of MCT1, involved the role of MCT1 in the nutritional support and functional operation of various tissues and organs in the whole body. In the second phase (2012–2016), there was a sharp growth of MCT1-related publications. The topic of tumors has long been the focus of medical researchers, and a number of studies exploring the associations between MCT1 and various human cancers were published in this period. Thus, this stage could be considered as the golden period of development for MCT1 research.

### 4.2. Analysis of Categories

For detecting and tracking the evolution of research related to MCT1, Citespace was applied to co-word category visualization about 851 publications related to MCT1 in the past 10 years. Research on MCT1 mainly focused on the notion that MCT1 could be a mediator on anaerobic glycolysis in tumor cells. Researchers analyzed the properties of MCT1 to consider the possibility as an effective therapeutic strategy. Therefore, research on cell biology, biochemistry, and molecular biology also increased rapidly. In the MCT1 research field, a multi-interdisciplinary system has been formed with oncology as the core subject.

### 4.3. Analysis of Leading Countries and Institutions

In terms of both the co-occurrence frequencies and betweenness centrality, USA could be viewed as an undisputed leading country in this research field. The developed countries with significant contributions such as USA and Germany were the major innovative forces in the research field about MCT1, equipped with widely admissive high-level scientific research and comprehensive networks of cooperation. Additionally, People’s Republic of China, as a model of developing country, has also made outstanding contributions in past 10 years.

According to the knowledge map of institutions, Universities as well as some research institutes represented the majority of institutions constructing studies on MCT1. Collaborative networks between research institutions were observed by CiteSpace. During the period ranging from 2008 to 2018, the most productive institutions were commonly located in Portugal, and they were University of Minho (40), University of Porto (34), and ICVS 3Bs PT Govt Associate Lab (31), respectively. The three most productive institution in Portugal formed intensive collaborations among each other and focused on the association between MCT1 and cancer therapy in unison.

The output comparison and network analyzing results between country and institution were not always consistent, mainly as Portuguese scholars had formed close cooperation systems about MCT1 research while scholars who come from productive countries such as USA and People’s Republic of China were scattered across a large number of different research institutions.

### 4.4. Analysis of Authors and Co-Cited Authors

Citespace helped to construct knowledge maps that provide information about the frequency and centrality of the authors and co-cited authors. The collaboration patterns between those core authors with strongest academic authority were also further analyzed. Additionally, as researchers published articles as a representative of their affiliations, the author analysis visualization results were similar to institutions. The top five prolific authors, including Baltazar (38), Pinheiro (27), Morris (18), Pellerin (14), and Sonveaux (14), tend to show a higher frequency of cooperation with others.

We also analyzed author co-citations by using Citespace. In those top five core strength researchers, Halestrap, Wilson, and Kirk all come from England, reflecting that England has displayed its world-leading level of scientific research in the MCT1 research field during last decade. Halestrap, the author with the largest number of citations, is based at Bristol University, and he contributed significantly to the existence of MCT1. Since then, exploding publications about MCT1 prompted researchers to seek for possibilities of MCT1 to be a therapy target on tumor cells [1]. Actually, his research was involved in many attributes of MCT1, and Halestrap has always played a major role in the MCT1 researching field. He studied MCT1 in an aspect of biochemistry and published many articles revealing the mechanism, structure, and properties of MCT1, making crucial contribution to MCT1 explicitation [16,17].

### 4.5. The Basic Mechanism and Characteristics of MCT1

Over the past few decades, the research in the MCT1 field has been constantly developing, which brought people further and further into MCT1. MCTs constituted a family of 14 members among which MCT1 facilitated the passive transport of monocarboxylates such as lactate, pyruvate, and ketone bodies together with protons across cell membranes [18]. MCT1 was widely distributed in various tissues and organs throughout the body, including normal tissues (e.g., skeletal muscles, adipose tissue, and brain) and pathological tissues (e.g., cancer tissue), and played an important role in many metabolic pathways, such as glycometabolism, lipogenesis, gluconeogenesis, and oxidative phosphorylate metabolic pathway. For instance, MCTs were the basic guarantee for the formation of a state of equilibrium, called “metabolic symbiosis” in tumor cells, where hypoxic cells exported lactate through MCT4, and oxygenated cells expressed MCT1 to import lactate to be used as a fuel [19]. A similar mechanism has been identified in the skeletal muscle [20,21] and brain [22,23]. Interactions with ancillary proteins, mainly basigin/CD147 and embigin/gp70, could adjust the correct expression, localization, and activity of MCTs [24,25,26]. Furthermore, with the changes of cell and tissue types and metabolic needs, MCT1 gene expression could also be regulated accordingly [18].

### 4.6. The Hotspots of MCT1 Research

According to the analysis of research hotspots, the focus of MCT1-related research has changed over time. However, the specific mechanism of MCT1’s participation in various metabolic pathways under normal physiological conditions (especially in skeletal muscle cells) and its expression in tumor cells under hypoxic conditions have always been hot topics in MCT1 research. In recent years, the special role of MCT1 in the aerobic glycolysis of tumor cells and its working mechanism in the metabolic process of adipose tissue have gradually become the new hotspots in this field. In the co-citation map (timeline view) of references, there were five articles (with a magenta ring) regarded as landmarks in the study of MCT1. These studies with different focuses were milestones in the MCT1-related study: (a) Suzuki et al. concluded that astrocyte-neuron lactate transport was necessary for long-term memory formation. This research has promoted a series of studies on the role of MCT1 in brain functions such as learning, memory, and language in rat brain models [27]. (b) In 2005, Murray et al. uncovered an undiscovered role for MCT1 in immune biology by adopting a chemistry-led target-identification strategy and suggested that MCT1 was a previously unknown target for immunosuppressive therapy [28]. (c) In 2008, Benton et al. found that PGC-1alpha increased skeletal muscle lactate uptake by increasing the expression of MCT1 but not MCT2 or MCT4 [29]. This study confirmed the correlation between MCT1 and skeletal muscle motion and provided a new idea for future studies on motor regulation.

### 4.7. The Frontiers of MCT1 Research

Burst keywords were regarded as indicators of research frontiers or emerging trends over time. Here, we listed three frontiers of MCT1 research as follows:

• Cancer and tumor cell

Metabolic cooperation has been identified in a variety of human cancers, and it relied on the expression and activity of MCTs at the cell membrane [30]. MCT1 and MCT4 were the most widely expressed subtypes of MCT in tumor cells among MCT family. A recent study pointed out that MCT1 was the most important isoform responsible for lactate transport across the plasma membrane in breast cancer, bladder cancer, non-small cell lung carcinomas (NSCLC), and ovarian carcinomas [31,32]. Many researchers have confirmed this viewpoint: Morais-Santos found that inhibition of MCT1 could impair lactate transport and the migration and invasion of breast cancer cells [33,34]. Sánchez-Tena indicated that butyrate-induced differentiation of colon cancer cells might be reduced by green tea phenolics via MCT1 [35]. Guo’s findings suggested that clinical outcomes might be influenced by SNPs (single nucleotide polymorphisms) in MCT1 and MCT2 genes, which could be used to predict the response to adjuvant chemotherapy in patients with NSCLC receiving surgical treatment in future study [36]. Therefore, MCT1 is of great clinical significance for the study of tumors, and studies in this field will go forward beyond all doubt.

• Apoptosis

MCT1 was involved in the uptake of lactate, and thus might be able to regulate the growth metabolism and apoptosis of the cells. Based on this functional characteristic, inhibiting the expression of MCT1 might affect the normal function and even survival rate of various cells in the body, so researchers started to explore MCT1 inhibitors few decades ago. CHC was employed for chemotherapy first [6], but it also inhibited mitochondrial pyruvate transport and glucose oxidation [37,38]. Then, AstraZeneca found a new class of specific and extremely high-affinity inhibitors of MCT1, which were originally thought to be potent inhibitors of T-lymphocyte proliferation as immunosuppressants [28,39,40]. A recent study indicated that inhibition of MCT1 by AZD3965 was a novel therapeutic approach for large B-cell lymphoma and Burkitt lymphoma [41]. These findings will inspire researchers to keep exploring more optimized MCT1 inhibitors and actively expand their use in other areas without doubt.

• Resistance

MCT1-mediated acid milieu promoted cancer malignance through multiple mechanisms, where drug resistance induction played a significant role [42]. On one hand, it interfered in the drug distribution and absorption due to physical pH change; on the other hand, MCT1 expression was positively correlated with enhanced glycolysis, and it participated in the process of lactate shuttle between cancer cells, which established metabolic symbiosis, facilitated drug resistance, and contributed to multidrug-resistant (MDR) phenotype of cells [43]. However, a recent study identified that specific proteins, such as MCT1, which was highly expressed on the surface of cancer cells, could transport deadly toxic substances into malignant cells, not the normal cells [44]. This distinction could be exploited to selectively kill cancer cells, and moreover, to help deal with the drug-resistance problem. Consequently, the specific application of MCT1 in the treatment of drug resistance of anti-cancer drugs needs further research. But, according to the results, there is no question that the exploration of the feasibility of MCT1 inhibitors as anti-cancer drugs will be the frontier and possible trend of research in this field in the future.

### 4.8. Strengths and Limitations

The research field of MCT1 could be regarded as an attractive and emerging field for the coming years, for cancers are still an intractable difficulty for humans. The special role of regulating lactate transport in the aerobic glycolysis of tumor cells made it an effective therapeutic strategy. Thus, it makes sense to carry out a visualization study on MCT1-related research. Besides, the analysis results constructed by CiteSpace could be observed more intuitively in comparison to simply reviewing existing literature. The method could, not only help us identify core researchers and collaboration networks among author, institution, country, but also characterize the historical and future development trends of research frontiers. Additionally, we could observe the crucial landmark papers with brief article information, including plenty of articles published in respected journals like *Science and Nature*.

There were also few main limitations considered for this bibliometric publication. Firstly, as information analyzed by Citespace was some specified introduction but not the full text due to its analysis feature, visualization figures could lack some crucial details when constructed. Besides, we only collected data from Web of Science and set the length of time as the most recent 10 years when exploring literature, which might lead to bias.

## 5. Conclusions

In this article, CiteSpace was applied to analyze the scientific output of MCT1 research in the world. As we know, this article is the first bibliometrics analysis about MCT1-related research using Citespace. Based on 851 publications published during the period of 2008–2018, we obtained a systematic and comprehensive overview of MCT1 research. The quantity of MCT1 research shows an ascendant trend over the past decade. The USA has been always acted as the dominant country over the past decade. A small cooperative group from several research institutions in Portugal tend to be the most prolific group recently, generating the major cluster of institutions. We also detected the research hotspots and frontiers by co-occurring and co-cited analysis. To draw conclusions about the results, the special role of MCT1 in human tumor cells have become hot points and focuses for scholars recently. Moreover, the burst-terms analysis indicated that identifying the clinical significance of MCT1 inhibitors as anti-cancer or immunosuppressive drugs and the possibility of drug-resistance formation are the frontier issues in this field. The development prospects of MCT1 research could be expected, and researchers should pay great attention to relevant studies.

## Figures and Tables

**Figure 1 ijerph-16-01091-f001:**
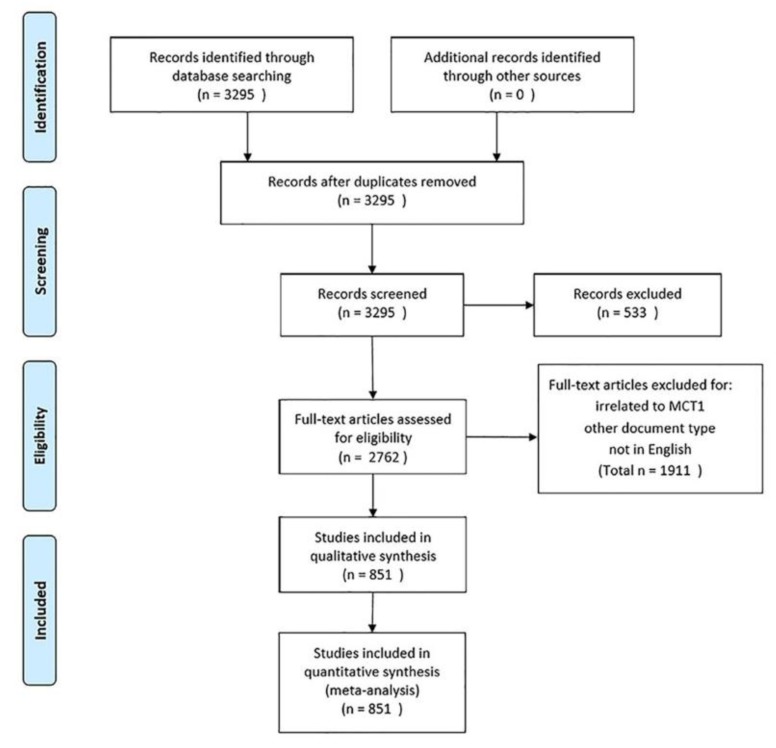
Flow diagram of study selection based on PRISMA 2009 guidelines.

**Figure 2 ijerph-16-01091-f002:**
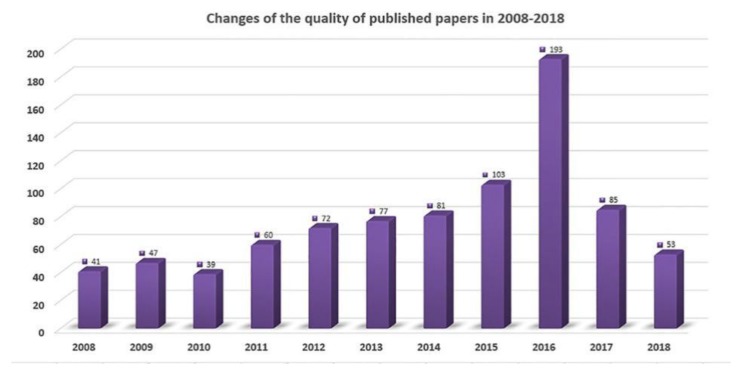
Changes of the quantity of annual publications on the research of monocarboxylate transport protein 1 (MCT1) indexed in the Web of Science during 2008–2018.

**Figure 3 ijerph-16-01091-f003:**
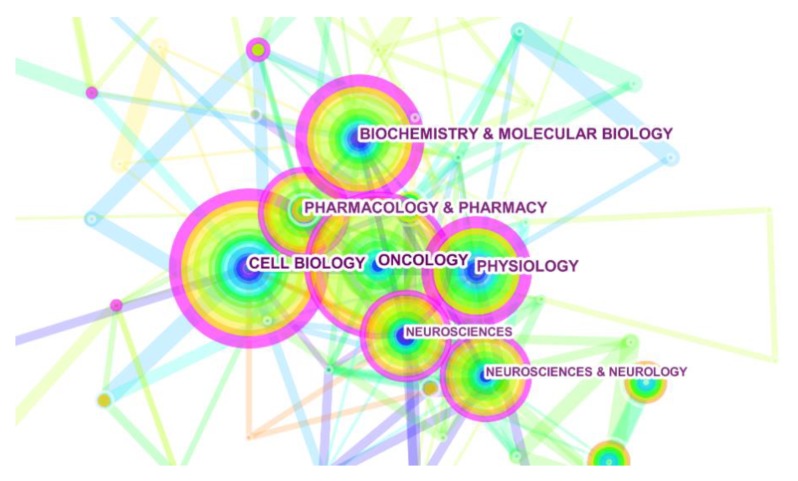
Visualization of research field. (Note: The size of a circle is in proportion to the amount of literature of the category, and the thickness of the lines is proportional to the relevance between different areas of research. The colors of rings of a circle are corresponding to the year. The purple rims of circles represent the high betweenness centralities).

**Figure 4 ijerph-16-01091-f004:**
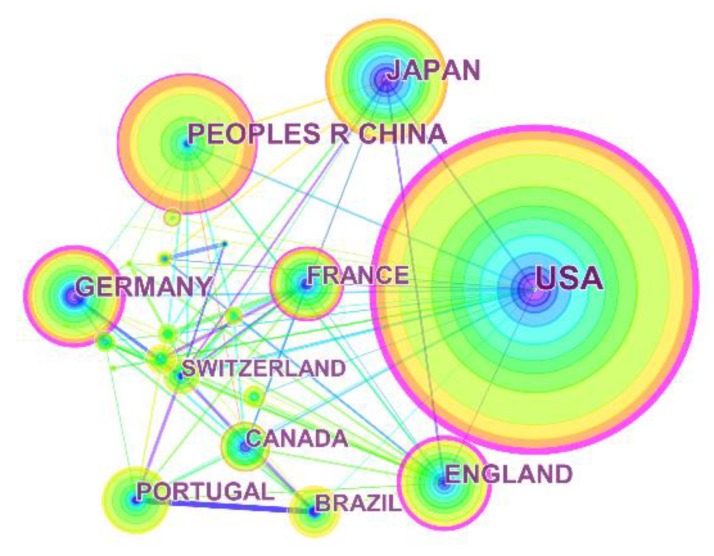
Visualization of leading countries. The country collaboration network of MCT1 constructed by CiteSpace has been observed with 36 nodes and 140 links, as 851 articles were contributed by 47 countries.

**Figure 5 ijerph-16-01091-f005:**
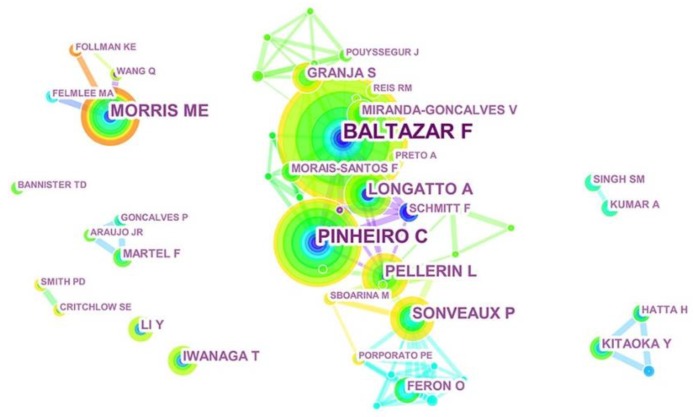
Map of the most productive authors contributed to MCT1 research with collaborative links. The author collaboration network of MCT1 constructed by CiteSpace was observed with 225 nodes and 436 links, as 851 articles were contributed by 225 authors. Each node represents an author, and each link reflects the collaboration network.

**Figure 6 ijerph-16-01091-f006:**
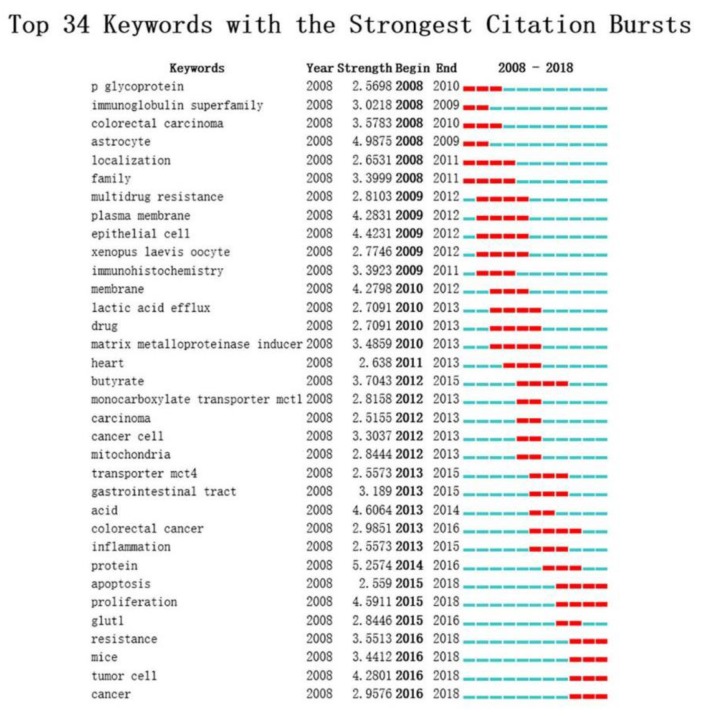
Top 34 keywords with the strongest citation bursts on the research of MCT1 indexed in the Web of Science during 2008–2018.

**Figure 7 ijerph-16-01091-f007:**
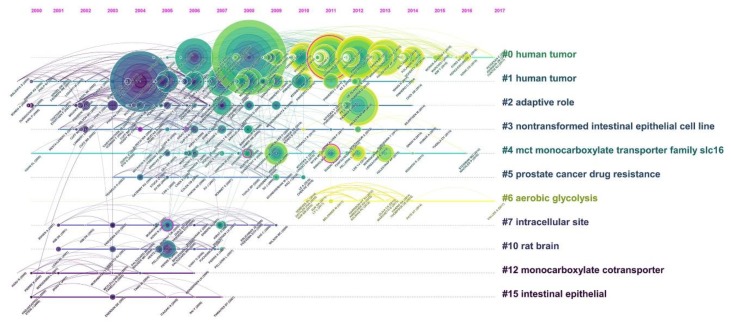
Co-citation map (timeline view) of references from publications on MCT1 research.

**Table 1 ijerph-16-01091-t001:** Top 10 most productive institutions that contributed to MCT1 research.

Rank	Count	Institution	Country	Continent	Research Focus
1	40	Univ Minho	Portugal	Europe	Targeting MCT1 as a treatment of cancer.
2	34	Univ Porto	Portugal	Europe	Targeting MCT1 as a treatment of cancer.
3	31	ICVS 3Bs PT Govt Associate Lab	Portugal	Europe	Targeting MCT1 as a treatment of cancer.
4	20	Univ Lausanne	Switzerland	Europe	MCT1 expression related to metabolism
5	19	Hokkaido Univ	Japan	Asia	Immunohistochemical localization and expression of MCT1
6	18	SUNY Buffalo	USA	North America	Targeting MCT1 as a treatment of GHB Overdose
7	18	Univ Copenhagen	Denmark	Europe	strength training on muscle and MCT1
8	17	Barretos Canc Hosp	Brazil	South America	MCT1 and anaerobic glycolysis in tumor cells
9	14	Univ Sao Paulo	Brazil	South America	Targeting MCT1 as a treatment of cancer.
10	13	Univ Manchester	England	Europe	MCT1 as functional biomarkers in cancer

**Table 2 ijerph-16-01091-t002:** Top 10 most active authors and co-cited authors contributed to MCT1 research.

Rank	Author	Country	Count	Co-Cited Author	Country	Citation
1	BALTAZAR F	Portugal	38	HALESTRAP AP	England	357
2	PINHEIRO C	Portugal	27	PINHEIRO C	Portugal	132
3	MORRIS ME	USA	18	SONVEAUX P	Belgium	130
4	PELLERIN L	Switzerland	14	WILSON MC	England	118
5	SONVEAUX P	Belgium	14	KIRK P	England	105
6	LONGATTO A	Brazil	14	DIMMER KS	Germany	98
7	BECKER HM	BECKER HM	10	ULLAH MS	England	97
8	GRANJA S	Portugal	9	BROOKS GA	USA	97
9	IWANAGA T	Japan	9	PELLERIN L	Switzerland	89
10	LI Y	USA	8	WARBURG O	Germany	82

**Table 3 ijerph-16-01091-t003:** Ranking of the frequency and centrality of keywords deprived from published articles on the research of MCT1 indexed in the Web of Science during 2008–2018.

Rank	Frequency	Keywords	Centrality	Keywords
1	224	expression	0.34	glycolysis
2	183	MCT	0.30	cd147
3	149	lactate	0.26	in vitro
4	140	metabolism	0.22	chain fatty acid
5	119	MCT1	0.19	central nervous system
6	90	cell	0.18	immunohistochemistry
7	79	hypoxia	0.16	immunoglobulin superfamily
8	71	gene expression	0.14	localization
9	69	Cd147	0.14	tumor
10	69	skeletal muscle	0.13	in vivo
11	63	glycolysis	0.11	MCT4
12	59	in vitro	0.11	Multidrug resistance
13	53	cancer	0.10	hypoxia
14	50	Cancer cell	0.10	gene expression
15	50	rat	0.10	brain

**Table 4 ijerph-16-01091-t004:** The top 10 co-cited journals in MCT1 research.

Rank	Co-Cited Journal	IF (2017)	Count
1	J BIOL CHEM	4.010	545
2	P NATL ACAD SCI USA	9.504	398
3	BIOCHEM J	3.857	393
4	CANCER RES	9.13	328
5	J PHYSIOL-LONDON	4.540	308
6	PLOS ONE	2.766	296
7	J CLIN INVEST	13.251	285
8	SCIENCE	41.058	284
9	NATURE	41.577	252
10	CELL	31.398	252

**Table 5 ijerph-16-01091-t005:** The analysis of clusters.

Cluster ID	Terms	Size	Sihouette	Mean (Year)
0	human tumor	104	0.786	2012
1	human tumor	49	0.827	2006
2	adaptive role line	48	0.921	2005
3	non-transformed intestinal epithelial cell l	36	0.933	2008
4	monocarboxylate transporter (MCT)	36	0.933	2006
6	prostate cancer drug resistance	22	0.892	2006
5	aerobic glycolysis	21	0.99	2011
7	intracellular site	17	0.978	2005
10	rat brain	13	0.974	2005
12	monocarboxylate cotransporter	9	0.985	2002
15	intestinal epithelial	6	0.988	2003

**Table 6 ijerph-16-01091-t006:** The top 10 co-cited references in MCT1 research.

Rank	Co-Cited Reference	Count	Main Findings
1	SONVEAUX P, 2008, J CLIN INVEST, V118, P3930	102	MCT1 inhibition has clinical antitumor potential.
2	HALESTRAP AP, 2004, PFLUGERS ARCH, V447, P619	81	An introduction to the SLC16 Gene Family.
3	HANAHAN D, 2011, CELL, V144, P646	68	The hallmarks of cancer comprise six biological capabilities, which constitute an organizing principle for rationalizing the complexities of neoplastic disease.
4	HALESTRAP AP, 2012, IUBMB LIFE, V64, P1	61	An introduction to the structure and functional characterization of the Monocarboxylate Transporter Family.
5	LE FR, 2011, P NATL ACAD SCI USA, V108, P16663	61	African monkeys are infected by Plasmodium falciparum nonhuman primate-specific strains.
6	PINHEIRO C, 2012, J BIOENERG BIOMEMBR, V44, P127	60	A literature review on the role of MCTs in cancer maintenance and aggressiveness in solid tumors in different locations.
7	HALESTRAP AP, 2012, IUBMB LIFE, V64, P109	57	An introduction to the role and regulation of the Monocarboxylate Transporter Family.
8	ULLAH MS, 2006, J BIOL CHEM, V281, P9030	55	The plasma membrane lactate transporter MCT4, but not MCT1, is up-regulated by hypoxia through a HIF-1alpha-dependent mechanism.
9	HALESTRAP AP, 2013, MOL ASPECTS MED, V34, P337	51	An introduction to the structure, role, and regulation of the Monocarboxylate Transporter Family in health and disease.
10	GALLAGHER SM, 2007, CANCER RES, V67, P4182	45	Monocarboxylate transporter 4 regulates maturation and trafficking of CD147 to the plasma membrane in the metastatic breast cancer cell line MDA-MB-231.

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
