# Peer review of "Research Trends and Hotspots Analysis Related to Monocarboxylate Transporter 1: A Study Based on Bibliometric Analysis"

_ijerph, 2019, doi:10.3390/ijerph16071091_

Round 1
Reviewer 1 Report
The authors analyzed the changes in the trend of the annual quantity of publications on MCT1 related during 2008-2018. Because MTC1 transporter may have an important role in cancer research the authors are evaluating if the interest in MCT1 has shifted or intensified in the last 10 year.The study was carefully designed and the manuscript is well written.
However a couple of sentences are not clear.
Title, The title refers to MCT1 one, however the authors seems to be talking interchangeable about also the other MCT’s show it should be explained in the test, why they are focusing on MCT1 and how the activity of MCT1 interacts with the others MCTs transporters.
Introduction;1) Page 1, line 45, Halestrap also reported that MCTs are expressed in the mitochondria and can be inhibited by 4-cin. 2) Page 2, line 48-49: "but also ingest extracellular lactic acid to supplement the glycolytic substrate". I think the authors should explain the process a little better and provide reference.
Author Response
Thanks for your suggestion. The main corrections in the paper and the responds to the reviewer’s comments are stated in the PDF file.

Reviewer 2 Report
The manuscript is a bibliometric analysis of Monocarboxylate transport protein 1. The article is easy to read and appropriately structured.
minor comments:
- line 39 PH must be changed to pH
- lines 42-49 Halestrap found that CHC prevent the absorption of pyruvate in erythrocytes, but this study was performed on 1974. The MCT1 was cloned in 1994.
- figures 3 and 4 were inverted
- more comprehensive captions are needed
Author Response

(The authors gave the same response as above.)

Reviewer 3 Report
Comments: I found the manuscript original and informative. This kind of studies are not very common but they can help authors to better identify the principal authors/institutions that are involved in the field of study and also the main findings.
Minor corrections:
Introduction, second paragraph: "but also ingest extracellular lactic acid to supplement the glycolytic substrate ...". Lactic acid is a supplement glycolysis? What is the "glycolytic substrate"?
There are also some typos and language incorrections throughout the text that need to be corrected.
Author Response

(The authors gave the same response as above.)
